# Dietary ARA, DHA, and Carbohydrate Ratios Affect the Immune Status of Gilthead Sea Bream Juveniles upon Bacterial Challenge

**DOI:** 10.3390/ani13111770

**Published:** 2023-05-26

**Authors:** Rui Magalhães, Nicole Martins, Filipa Fontinha, Rolf Erick Olsen, Claudia Reis Serra, Helena Peres, Aires Oliva-Teles

**Affiliations:** 1Interdisciplinary Centre of Marine and Environmental Research (CIIMAR), University of Porto, Terminal de Cruzeiros do Porto de Leixões, Av. General Norton de Matos s/n, 4450-208 Matosinhos, Portugal; 2Departamento de Biologia, Faculdade de Ciências, University of Porto, Rua do Campo Alegre s/n, Ed. FC4, 4169-007 Porto, Portugal; 3Department of Biology, Norwegian University of Science and Technology, N-7491 Trondheim, Norway; rolf.e.olsen@ntnu.no

**Keywords:** essential fatty acids, bacterial challenge, hematology, innate immune system, gene expression

## Abstract

**Simple Summary:**

As a consequence of the increased use of plant ingredients in marine fish diets, there is a trend for a reduction in long-chain polyunsaturated fatty acid content and an increase in carbohydrates in aquafeeds. Thus, it is important to understand the ways in which such dietary modifications impact the health status and disease resistance of fish. Considering this, we evaluated the effects of high dietary inclusions of arachidonic acid (ARA) or docosahexaenoic acid (DHA) together with high or low inclusion of digestible carbohydrates (CHO) before and after a bacterial challenge with killed *Photobacterium damselae* subsp. *piscicida* (*Phdp*) on selected immune parameters. Our results showed that high dietary ARA levels improved the fish immune response to a bacterial challenge. The dietary CHO provides important energy to promote a fast response of the gilthead sea bream immune system.

**Abstract:**

This study aims to assess the effects of different dietary n-6/n-3 long-chain polyunsaturated fatty acid ratios and CHO content in the immune response of gilthead seabream. For that purpose, gilthead sea bream juveniles (initial body weight = 47.5 g) were fed for 84 days with four isoproteic (47% crude protein) and isolipidic (18% crude lipids) diets with high (20%) or low (5%) level of gelatinized starch (HS or LS diets, respectively) and included approximately 2.4% ARA or DHA. At the end of the trial, the DHA-enriched groups presented increased red blood cell (RBC) count, hemoglobin, plasmatic nitric oxide (NO) content, and antiprotease and alternative complement activities. The ARA groups had increased thrombocyte count, and plasmatic bactericidal activity against *Vibrio anguillarum* was lower in the fish fed the ARA/LS diet. After the feeding trial, the fish were challenged with an intraperitoneal injection (i.p.) of killed *Photobacterium damselae* subsp. *piscicida* (*Phdp*) and sampled at 4 and 24 h after the challenge. At 4 h after i.p., the ARA groups presented increased plasma total immunoglobulins (Ig) and bactericidal activity against *V. anguillarum*. In addition, the fish fed the ARA/LS diet presented lower white blood cell (WBC) and alternative complement activity. At 24 h after i.p., the ARA groups presented increased RBC, WBC, and thrombocyte numbers, total IG, plasma peroxidase activity, and casp3 expression in the distal intestine. The HS groups presented increased plasma NO content and bactericidal activity against *Phdp* and decreased protease, antiprotease activity, and bactericidal activity against *V. anguillarum*. In conclusion, high dietary DHA levels seemed to improve the immune status of unchallenged gilthead sea bream juveniles, while high dietary ARA levels improved the fish immune response to a bacterial challenge. The energy provided by dietary starch seems to be important to promote a fast response by the fish immune system after a challenge.

## 1. Introduction

The utilization of vegetable and animal oils to replace fish oil (FO) in commercial aquafeeds for marine fish has been receiving a great deal of attention because the majority of marine fish have limited capacity to biosynthesize linoleic (18:2n-6) and α-linolenic (18:3n-3) acids into long-chain polyunsaturated fatty acids (LC-PUFA) such as arachidonic (ARA; 20:4n-6) and docosahexaenoic (DHA; 22:6n-3) acids due to the absence or low activity of ∆5-desaturase and elongases (Elovl_2_) [1].

ARA and DHA (also EPA) are fatty acids with opposite properties regarding inflammatory and anti-inflammatory responses and inflammation resolution [2]. Thus, for an adequate inflammatory response, diets must fulfill the requirements for these essential fatty acids [3]. 

ARA is a substrate for the synthesis of eicosanoids such as prostaglandin E2 (PGE2) and leukotriene B4 (LTB4) which promote leucocyte chemotaxis, reactive oxygen species formation, and other pro-inflammatory effects such as vascular permeability and vasodilatation [2,4,5]. The production of inflammatory cytokines such as TNF-α, IL-1β, and IL-6 is also associated with ARA-related eicosanoids [2]. ARA preferential incorporation in leucocytes was described in different fish species [6,7] and the immune system modulation by ARA has been highlighted in different studies [8,9,10,11]. ARA also participates in the assembling and activation of NADPH oxidase, thus controlling respiratory bursts, and in the anti-microbial function of infiltrating neutrophils and other leucocytes [4]. The importance of ARA in the immune system is also associated with the capacity to modify protein-encoding genes that directly impact immune-related transcription factors such as nuclear factor kappa B (NF*_k_*B) [5].

The modulation of the immune system is also induced by n-3 LC-PUFA, namely by EPA and DHA, which compete with ARA as substrates for cyclooxygenase 2 (COX_2_) and 5-lipoxygenase (5-LOX) for producing eicosanoids with less inflammatory potential than those produced through ARA [12,13]. Thus, n-3 LC-PUFA counteracts ARA inflammatory responses. Moreover, both EPA and DHA originate resolvins which resolve the inflammatory process, and DHA also generates docosanoids with anti-inflammatory and immunomodulatory properties [13].

In fish, DHA seems to be more effective than EPA as an immunomodulator, and dietary DHA/EPA ratios higher than one have been recommended for several marine species [14,15,16,17].

In gilthead sea bream, several studies have already evaluated the effects of different dietary essential fatty acid (EFA) profiles on the immunological status of fish under unchallenged and challenging conditions since this species has limited capacity for LC-PUFA biosynthesis [18]. For instance, a balanced dietary ARA: EPA+DHA ratio (1% ARA; 0.4% EPA; 0.4% DHA) increased the monocyte numbers, alternative complement activity, and bactericidal activity against *Photobacterium damselae* subsp. *piscicida* (*Phdp*) in fish compared to the fish fed an ARA-rich diet (2%) or n-3 LC-PUFA-rich diets (0.6%:0.6% EPA: DHA or 0.3%:1.5% EPA: DHA) [19]. However, after being intraperitoneally injected with killed *Phdp*, the fish fed a DHA-rich diet (0.3%:1.5% EPA: DHA) presented higher alternative complement activity and intestinal expression of pro-inflammatory genes while the fish fed an ARA-rich diet (2%) had higher plasma bactericidal activity against *Phdp* [20].

Carbohydrates (CHO) are a low-cost energy and carbon source for fish. They are incorporated in aquafeeds to decrease feed costs and to spare protein use for growth [21]. However, carnivorous fish have a limited ability to digest and utilize dietary CHO [22]. Dietary CHO are glycogenesis and lipogenesis enhancers [23,24] and were also shown to up-regulate desaturases and elongases in salmonids [25,26] and gilthead sea bream [23]. Thus, dietary CHO may contribute to modifying the available fatty acid pool, ultimately leading to changes in the eicosanoid-associated immune response. Studies about the role of dietary CHO in fish health are scarce, but the available evidence shows that they may affect the innate immune response of fish [27,28,29,30,31]. The type of carbohydrate present in the diet is also relevant. In mice, dietary glucose showed to increase lymphocyte B numbers at the early stages of the lymphopoieses compared to fructose by regulating glycolysis and oxidative phosphorylation and protect from apoptosis through mTOR activation [32]. In this study, high digestible carbohydrates increased IgG response in the plasma of immunized animals [32].

Gilthead sea bream is an important species for European aquaculture [33] and, although it is a carnivorous species, it efficiently uses diets with 20% digestible CHO [34]. The n-3 LC-PUFA requirements of gilthead sea bream juveniles were estimated to be 0.9 or 1.9%, depending on whether the DHA: EPA ratio was 1 or 0.5, respectively [35,36]. This highlights the relevance of DHA compared to EPA as EFA for this species. ARA requirements for gilthead sea bream are still not known, but they seem to be very low. A previous study showed that dietary levels ranging from 0.03% to 1.7% did not affect growth performance and feed utilization [37].

The objective of this study was to evaluate the effects of dietary ARA, DHA, and CHO levels on selected immune parameters of gilthead sea bream before and after being submitted to a bacterial challenge.

## 2. Materials and Methods

### 2.1. Experimental Diets

The experimental diets were formulated as described in Magalhães et al. [38]. Briefly, four diets were designed to be isoproteic and isolipidic (47% crude protein; 18% crude lipids) and to include approximately 2.4% ARA or DHA of the diet with low (5%; LS) or high starch (20%; HS). The diets were dry pelleted in a pellet mill (California Pellet Mill, CPM, Crawfordsville, IN, USA), dried at 40 °C for 48 h, and stored at −4 °C until utilization. Table 1 and Table 2 present a summary of the most relevant data.

### 2.2. Bacteria Inoculum Preparation

Formalin-killed bacterin was prepared with the virulent strain Lg_H41/01_ of *P. damselae* subsp. *piscicida (Phdp)* according to Diaz-Rosales et al. [39]. Initially, bacteria were cultured on tryptic soy agar supplemented with 1.5% NaCl (*w*/*v*) (TSAs) for 48 h at 22 °C. Then, one colony was transferred to the tubes containing 5 mL of tryptic soy broth supplemented with 1.5% NaCl (TSBs) and incubated for 18 h at 22 °C. The next day, 50 μL aliquot of the culture was inoculated in flasks with 50 mL TSBs and incubated at 22 °C for 24 h with continuous shaking. The bacterial culture was then adjusted to 1 × 10^8^ CFU mL^−1^ and killed with formaldehyde (1% final concentration) for 24 h. The formalin-killed bacterin was collected by centrifugation at 6000× *g* for 30 min at 4 °C, washed 3 times in phosphate-buffered saline (PBS), and resuspended in the same volume of PBS (50 mL). The efficacy of formalin treatment was confirmed by plating on TSA plates and incubating for 2 days at 22 °C.

### 2.3. Challenge Trial

The experiment was approved by the CIIMAR ethical committee for Managing Animal Welfare (ORBEA; reference ORBEA_CIIMAR_30_2019) in compliance with the European Union directive 2010/63/EU and the Portuguese Law (DL 113/2013). The feeding trial was performed as described in Magalhães et al. [38], in a thermo-regulated water system (23 ± 1 °C) equipped with 100 L capacity tanks. The trial was performed with triplicate groups of gilthead sea bream (*Sparus aurata*) with 47.5 g initial body weight, and the fish were fed by hand until apparent visual satiation for 12 weeks. Then, the blood from 3 fish per tank was collected from the caudal vein with heparinized syringes. Thereafter, 24 fish per dietary treatment (135 g mean body weight) were intraperitoneally injected (i.p.) with 0.6 mL of the killed *Phdp* (challenged group) or with 0.6 mL PBS (sham group). The 12 fish from the challenged and sham groups were reallocated into two different tanks according to stimuli. At 4 and 24 h after intraperitoneal injection, the blood from the six fish previously injected with the killed *Phdp* and with PBS was collected from the caudal vein. Then, the fish were euthanized by decapitation and a portion of the distal intestine (DI, distinguished from the mid-intestine by an enlarged diameter and darker mucosa) was sampled for gene expression. DI samples were maintained in an RNAlater solution (1:10) for 24 h at 4 °C and then stored at −80 °C until analyzed.

After blood collection, an aliquot was used to assess haematological parameters, and the remaining blood was immediately centrifuged at 6800× *g* for 10 min. Plasma was collected and frozen at −80 °C until immunological parameters analysis.

### 2.4. Haematological Parameters

Blood smears were made immediately after blood collection, air-dried, fixated with formol–ethanol (3.7% formaldehyde in absolute ethanol), and stained with Wright’s stain (Haemacolor; Merck). Then, the glass slides were examined (1000×) under oil immersion, and at least 200 leucocytes were counted and classified as lymphocytes, thrombocytes, monocytes, and neutrophils. Neutrophil identification was performed by the detection of peroxidase activity according to Afonso et al. [40]. The absolute value (×10^4^ μL^−1^) of each cell type was subsequently calculated.

Blood parameters were assessed according to Peres et al. [41]. Total red (RBC) and white (WBC) blood cell counts were performed using a hemocytometer. Fresh heparinized blood was spun for haematocrit (Ht) using microhematocrit tubes (10,000× *g* for 10 min at room temperature). Drabkin’s solution was used for haemoglobin determination (HB; SPINREACT kit, ref. 1001230, Girona, Spain). The mean corpuscular volume (MCV), mean corpuscular haemoglobin (MCH), and mean corpuscular haemoglobin concentration (MCHC) were calculated as follows:MCV (μm^3^) = (Ht/RBC) × 10,
MCH (pg cell^−1^) = (HB/RBC) × 10,
MCHC (g 100 mL^−1^) = (HB/Ht) × 100.

### 2.5. Plasma Immune Parameters

Protease activity was determined by the azocasein hydrolysis method and antiprotease activity was assessed by the ability of plasma to inhibit trypsin activity according to Machado et al. [42].

Total peroxidase activity was assessed as described by Quade and Roth [43], and one unit of peroxidase activity was defined as that results in an absorbance change of 1 OD (units mL^−1^ of plasma).

Total nitrite plus nitrate content was quantified using a Nitrate/Nitrite colorimetric kit (Roche Diagnostics GmbH, Mannheim, Germany, ref: 11746081001) adjusted for 96-well microplates. As both compounds are oxidative metabolites of endogenously produced nitric oxide (NO), they were used to quantify the NO content in plasma.

Total immunoglobulins (Ig) were determined following Siwicki and Anderson [44], based on the measurement of plasma total protein content using the Pierce™ BCA Protein Assay Kit (Thermo Scientific™, ref: 23225, Rockford, IL, USA) before and after the precipitation of Ig molecules by using a 12% solution of polyethylene glycol. Total plasma Ig content was determined by the difference between protein content.

Alternative pathway complement activity was measured according to Sunyer and Tort [45]. The haemolysis degree (Y) was estimated by plotting Y(1-Y)^−1^ on a log–log scale graph against the lysis curve for each sample. The volume of plasma producing (50%) hemolysis (ACH_50_) was estimated and the number of ACH_50_ units mL^−1^ was calculated for each sample.

Bactericidal activity was assessed against two opportunist marine pathogenic bacteria (*Vibrio anguillarum* and *Photobacterium damselae* subsp. *piscicida* (*Phdp*)) according to Graham et al. [46]. Total bactericidal activity was provided as the percentage of killed bacteria, calculated from the differences between the percentage of bacteria that survived and the positive control (100% living bacteria).

### 2.6. Gene Expression

The evaluation of mRNA levels was measured on the DI according to Magalhães et al. [38]. Briefly, RNA extraction was performed using a TRIzol reagent (Direct-zolTM RNA Miniprep, Zymo Research), and for the cDNA synthesis, the NZY First-Strand cDNA Synthesis Kit (NZYTech, MB12501, Lisbon, Portugal) was used following the manufacturer’s recommendations. Gene expression was quantified by real-time PCR (CFX ConnectTM Real-Time System, Bio-rad, Hercules, CA, USA). The primers were obtained in the literature except for the sequences for prostaglandin E2 receptors EP_2_, EP_3,_ and Ep_4_ that were obtained from Genebank (Table 3). Primer efficiency was determined according to Pfaffl [47]. The normalization of the target gene was performed using the elongation factor 1 α (*ef1α*) and 18 s ribosomal RNA (*18s*) as a reference gene. Expression levels are provided as relative quantification of the target gene calculated by the ratio between copy numbers of the gene of interest and the geometric mean of copy numbers of reference genes *ef1α* and *18s* following the method described by Vandesompele et al. [48].

### 2.7. Statistical Analysis

Data of the hematological and immune status of gilthead sea bream juveniles after the feeding trial are presented as means, and data of bacterial challenge are presented as means of the fold change relative to the mean value of the PBS-injected fish (defined as 1) at the same sampling time ± pooled standard error (PSE) or standard error of the mean (SE). The normality and homogeneity of variances were assessed and log-transformed when needed. Data were analyzed by two-way ANOVA, with EFA or CHO levels as fixed factors. If interactions were detected, a one-way ANOVA was performed for each factor. The null hypothesis was rejected with a significance level of 0.05.

## 3. Results

The growth performance of the fish fed the experimental diets and results were presented elsewhere [38]. Briefly, growth performance was not affected by dietary treatments, indicating that the dietary n-3 LC-PUFA requirements of gilthead seabream juveniles were fulfilled with a diet including 0.5% EPA+DHA, at least in the presence of high levels of ARA. Nevertheless, a trend towards higher feed intake and lower growth performance led to a significant decrease in the feed efficiency of the fish fed with the LS diets.

### 3.1. Hematological and Immune Status at the End of the Feeding Trial

At the end of the feeding trial, the fish RBC numbers and the HB content were higher in the fish fed DHA-rich diets (Table 4). No further differences between groups were observed in the other hematological parameters measured, except for thrombocyte numbers, which were lower in fish fed the ARA/LS diet.

Plasma nitric oxide, antiprotease, and alternative complement activities were higher in the fish fed the DHA-rich diets, while protease activity was higher in the fish fed the ARA-rich diets (Table 5). Plasmatic bactericidal activity against *V. anguillarum* was lower in the fish fed the ARA/LS diet (Table 5).

Distal intestine expressions of lipoxygenase (*5-lox*), cyclooxygenase 2 (*cox*_2_), interleukin 1β (*il-1β*), tumor necrosis factor α (*tnfα*), immunoglobulin M heavy chain (*igm*), major histocompatibility complex II (*mhc-II*), interleukin 10 (*il-10*), prostaglandin E2 receptor *ep_2_* and *ep_4_* were not affected by the dietary treatments (Figure 1). The ARA diets induced the expression of prostaglandin E2 receptor *ep*_3_ while caspase3 (*casp3*) expression was induced in the fish fed the DHA/HS diet.

### 3.2. Hematological and Immune Status at 4 h after Challenge

At 4 h after the challenge with the killed *Phdp,* RBC count was higher in the fish fed with the DHA-rich diet, but only within the HS group (Table 6). The HT and thrombocyte numbers were higher while MCHC was lower in the fish fed the DHA-rich diets. The WBC numbers were lower in the fish fed the ARA/LS diet than in the other groups.

Total Ig and plasmatic bactericidal activity against *V. anguillarum* were higher in the fish fed the ARA-rich diets than the DHA-rich diets (Table 7). Alternative complement activity was much lower in the fish fed the ARA/LS diet than in the other groups, and it was higher in the fish fed with the ARA/HS diet than the DHA/HS diet.

The expression of 5-*lox*, *cox_2_*, *il-1β*, *tnfα*, *igm*, mhc-*II*, *il-10*, and *casp3* in the distal intestine was not affected by dietary treatment while the expression of *ep*_2_ and *ep*_4_ was higher in the fish fed the HS diets (Figure 2).

### 3.3. Hematological and Immune Status at 24 h after the Challenge

At 24 h after the challenge with the killed *Phdp,* RBC, WBC, and thrombocyte numbers were higher in the fish fed the ARA-rich diets, while the HB and MCV values were higher in the LS groups (Table 8). The HT was higher in the fish fed the DHA/LS diet than in the other groups. In the ARA groups, the MCHC was higher in the fish fed the LS diet, while the opposite was observed within the DHA groups.

Plasma total Ig levels and peroxidase activity were higher in the fish fed the ARA-rich diets (Table 9). Nitric oxide and bactericidal activity against *Phdp* were higher in the fish fed the HS diets while protease, antiprotease, and bactericidal activity against *V. anguillarum* were higher in the fish fed the LS diets.

The expression of *casp3* in the distal intestine was higher in the fish fed the ARA-rich diets (Figure 3). No statistical differences were found between dietary treatments in *mhc-II* expression, although a significant interaction between the EFA ratio and starch level was noticed.

## 4. Discussion

In this study, major effects on the immune parameters assessed were related to the dietary fatty acid ratios, while dietary CHO effects were less pronounced and occurred mainly after the bacterial challenge. In the following sections, the effects observed before and after the bacteria challenge trial are discussed separately.

### 4.1. Dietary Effects on Hematological and Immune Status at the End of the Feeding Trial

At the end of the feeding trial, hematological parameters were unaffected by diet composition, except for the RBC counts and HB content that were increased in the fish fed the DHA-rich diets. In a previous study on gilthead sea bream juveniles fed diets with different ARA, EPA, and DHA ratios, no differences in RBC counts and HB concentrations were observed [19]. However, the dietary ARA:DHA ratios tested in the present study were higher and this may contribute to explaining the differences observed.

Plasma innate immune parameters were affected by the dietary ARA:DHA ratios, but not by dietary CHO level, except for the bacterial activity against *V. anguillarum* that was decreased in the fish fed low-CHO diets, but only in the high ARA group.

The lower levels of NO, antiprotease and alternative complement activities in the fish fed the high ARA diets may be related to a possible n-3 LC-PUFA EFA deficiency. Previously, in gilthead sea bream, the depletion of alternative complement activity has been associated with the n-3 LC-PUFA dietary deficiency [49,50], and a balanced EFA diet (1.0% ARA, 0.4% EPA, and 0.4% DHA) increased the plasmatic alternative complement activity and bactericidal activity against *Photobacterium damselae* compared to the fish fed an ARA-rich diet (2%) or a diet rich in n-3 LC-PUFA (0.3% EPA and 1.5% DHA) [19]. In this trial, diets included only circa 0.5% n-3 EFA (EPA+DHA), which is lower than the estimated n-3 EFA requirement for this species [35,36]. Although this amount seemed sufficient to promote an adequate growth performance [38], it might be insufficient to meet adequate immune responses. Nevertheless, the slightly but significantly higher plasma protease activity in the ARA groups and the lack of effects in the other innate immune parameters analyzed suggest that the overall immune response was not compromised. This seems to be confirmed by the immune-related gene expression in the distal intestine that, except for *ep_3_* expression, was also not affected by diet EFA composition. Previously, it was also observed in this species that distal intestinal immune-related gene expression was not modulated by the dietary ARA, EPA, or DHA levels [19].

ARA is the precursor of PGE_2_, and upregulation of prostaglandin E2 receptor *ep_3_* in the fish fed with ARA-rich diets may indicate an adaptation of the immunomodulatory response related to the high ARA availability. Indeed, in European seabass juveniles, it was also observed that plasma prostaglandins levels increased in response to increasing dietary ARA levels [11].

As for the plasma humoral parameters, dietary CHO did not affect immune gene expression, except for *casp3* which was downregulated in the fish fed the low-CHO diets, but only in the DHA group, possibly because of the lower digestible energy and ARA/DHA ratio of this diet.

### 4.2. Dietary Effects on Hematological and Immune Status after Challenge with Killed Phdp

At 4 h after the challenge, the RBC count was higher in the fish fed the DHA/HS diet compared to the fish fed the ARA/HS diet, while at 24 h after the challenge, the RBC count was higher both in the fish fed the HS diets and in the fish fed the ARA-rich diets. Also in this species, it was previously observed that the fish fed a DHA-rich diet (1.5%) presented higher RBC count than the fish fed an ARA-rich diet (2%) at 4 h after being challenged with killed *Phdp* but, contrary to the results of the present trial, no differences due to the dietary EFA was observed at 24 h after the challenge [20].

The WBC count at 4 h after the challenge was higher in the fish fed the DHA-rich diets, but the opposite was observed at 24 h after the challenge. The differences in WBC counts were mainly due to the thrombocyte number variation, as no differences in neutrophil and monocyte numbers were observed at the two sampling times, and the lymphocyte count was only affected at 4 h after the challenge in the fish fed the ARA/LS diet.

This response difference in the WBC count at the two sampling times may be related to a delayed response in the fish fed the ARA diets, as was also the case for the RBC count. Previously, 24 h after the challenge with killed *Phdp*, the gilthead sea bream juveniles fed diets with low ARA and a balanced DHA:EPA ratio (1.2% EPA+DHA; DHA:EPA= 1:1) had higher WBC and thrombocytes counts than the fish fed DHA-rich diets (1.5%; DHA:EPA ratio = 5:1; [20]) but similar to the fish fed ARA rich diets (2% ARA/0.3% EPA+DHA; 1% ARA/0.8% EPA+DHA).

Nitric oxide response and bactericidal activity were not affected by the dietary EFA but were increased in the fish fed the HS diets, and this effect was statistically significant at 24 h after the challenge. On the contrary, protease and antiprotease activities were higher in the fish fed the LS diets and were also not affected by the dietary EFA.

Overall, this seems to indicate that dietary CHO contributes to better plasmatic immune homeostasis in fish submitted to a bacterial challenge, probably by more efficacy in providing the digestible energy required for an adequate response. These results induced by dietary starch (higher digestible energy) could be associated with better performance of the WBC, cells with high energy demand, since no difference in numbers was noticed. In mice, dietary glucose showed to increase lymphocyte B survival at the early stages of the lymphopoieses by regulating glycolysis and oxidative phosphorylation and protecting from apoptosis through mTOR activation, and high-CHO diets increased IgG response in the plasma of immunized animals [32]. There is not much information available on the effect of dietary CHO on fish plasma immune parameters, and the available data are somehow contradictory. For instance, in juvenile golden pompano (*Trachinotus ovatus*), dietary inclusions of up to 16.8% CHO enhanced the plasmatic non-specific immune parameters, but further inclusion (up to 28%) led to the opposite effect [27]. In addition, in black carp (*Mylopharyngodon piceus*), a deficiency or an excess of dietary CHO decreased plasmatic immune parameters, ultimately leading to diminished resistance against infection with *Aeromonas hydrophilla* [28]. In 500 g Atlantic salmon, haemolitic activity was negatively correlated with dietary CHO (0 to 30% DM), and survival after a challenge with *Aeromonas salmonicida* was reduced with more than 10% dietary CHO [31]. In tilapia, 18% of dietary corn starch increased plasma lysozyme activity and reduced mortality after a challenge against *Aeromonas hydrophila* compared with fish fed 36% of dietary corn starch [30]. The results of the different studies showed the importance of an adequate level of dietary CHO to the fish immune system.

Regarding dietary EFA, dietary ARA increased total immunoglobulins and bactericidal activity against *V. anguillarum* at 4 h after the challenge and total immunoglobulins and peroxidase at 24 h after the challenge, while dietary DHA did not improve any of the plasma immune parameters response. Alternative complement activity was also highly increased in the fish fed the ARA/HS diet but was highly decreased in the ARA/LS fish, suggesting a strong effect due to the lower availability of CHO in this group. The overall response to the dietary EFA was somehow expected, as ARA is the precursor of the inflammatory response, while DHA is associated with the resolution of the inflammatory response [2]. As the sampling times were too short after the bacteria insult, fish would still be developing an inflammatory response. The role of ARA as an immunomodulator in fish was previously highlighted by other authors with similar significant improvement of non-specific immune parameters associated with dietary ARA supplementation [9,10,11].

Eicosanoid production through *cox_2_* and *5-lox* is directly responsible for the regulation and activation of biochemical cascades within the immune system [13]. However, in the present study, neither *cox_2_* nor *5-lox* expression in the distal intestine was affected in response to the dietary composition. Thus, the lack of variation in the expression of most immune-related genes measured was to be expected. Contrarily, in a previous study with gilthead sea bream also challenged with killed *phdp*, DHA-rich diets (1.5%; DHA: EPA ratio = 5:1) induced the expression in the distal intestine of *cox_2_* and *il-1β* at 4 and 24 h and of *tnfα* at 4 h after the challenge [20].

In the present study, *casp3* expression in the distal intestine was induced at 24 h after challenge in the fish fed the ARA diets. Previous studies also showed an increased *casp3* production in fish in response to infection by bacteria. For instance, in striped murrel (*Channa striatus*)*, casp3* gene expression was upregulated at 24 h after inoculation with *Aeromonas hydrophila* [51] and in tongue sole (*Cynoglossus semilaevis*) *casp3* was upregulated at 12 h following the *Edwardsiella tarda* infection [52]. Thus, the present results further confirm the importance of *casp3* in the mechanism of response to infection by bacteria and the positive effect of ARA in inducing this response.

The upregulated expression of *ep_2_* and *ep_4_*, the PGE_2_ receptors, in the fish fed the HS diets 4 h after the challenge further confirms the importance of dietary CHO to provide energy for a fast immune response. An increase in immune-related gene expression was also observed in black-carp-fed diets with adequate levels of dietary CHO [29].

## 5. Conclusions

Overall, this study shows that high dietary ARA was more effective than DHA in promoting short-term immune response in gilthead sea bream challenged with killed *Phdp*, and that adequate dietary carbohydrates are important to provide available energy for the fast immune response of the fish. In the future, it will be interesting to test the effect of diets with high levels of ARA and DHA (1:1) and carbohydrates to assess whether there is a synergetic or an inhibitory effect on gene expression and overall immune response.

## Figures and Tables

**Figure 1 animals-13-01770-f001:**
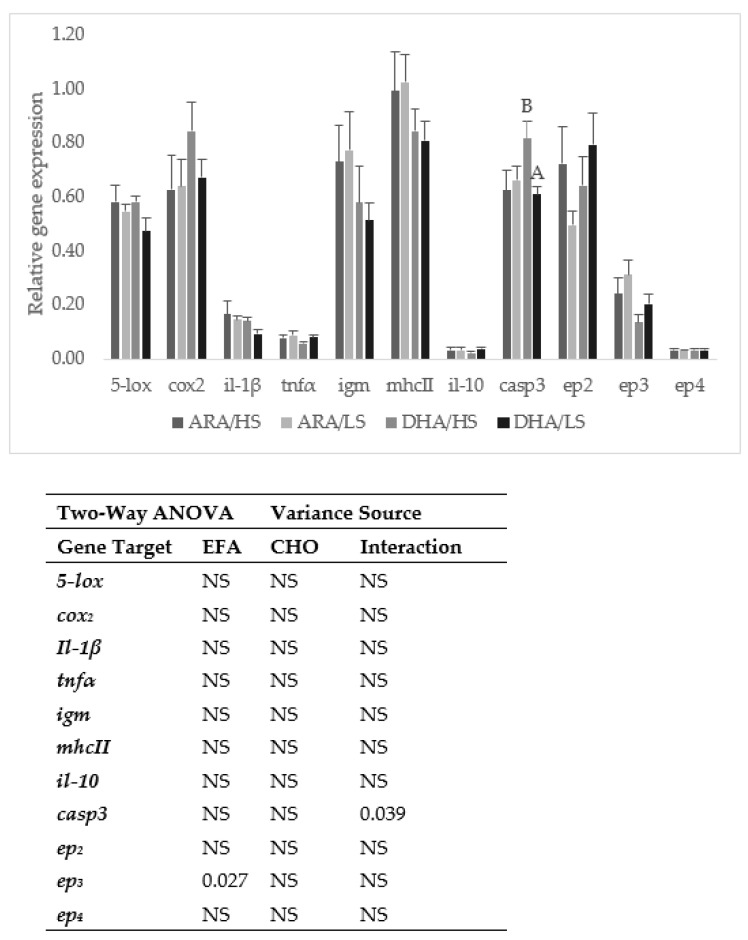
Intestinal expression levels of lipoxygenase (*5-lox*), cyclooxygenase 2 (*cox*_2_), interleukin 1β (*il-1_β_*), tumor necrosis factor α (*tnf_α_*), immunoglobulin M heavy chain (*igm*), major histocompatibility complex II (*mhc-II*), interleukin 10 (*il-10*), caspase 3 (*casp*_3_), and prostaglandin E2 receptor *ep*_2_, *ep*_3,_ and *ep*_4_ in gilthead sea bream fed the experimental diets for 84 days. Sample expressions were calibrated with the sample with a higher expression value and then normalized with elongation factor 1-alpha (*EF1α*) and ribosomal RNA 18S (*18S*) expressed transcripts. Values presented as means (n = 9) and standard error of the mean (SE); Two-way ANOVA: NS: non-significant (*p* ≥ 0.05). A different capital letter indicates differences between CHO levels within each EFA.

**Figure 2 animals-13-01770-f002:**
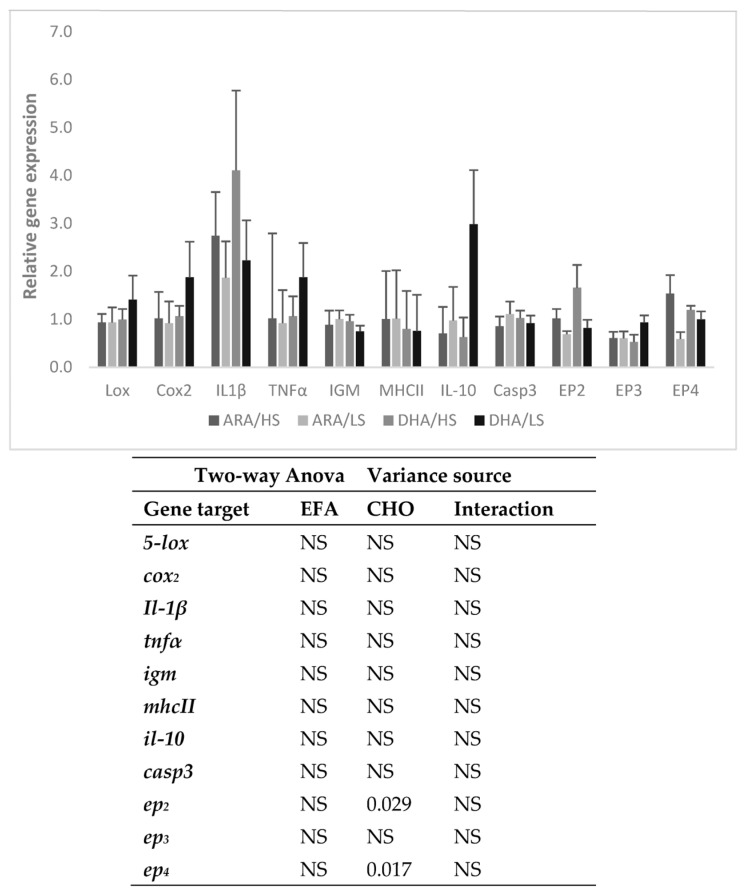
Intestinal expression levels of lipoxygenase (*5-lox*), cyclooxygenase 2 (*cox*_2_), interleukin 1β (*il-1_β_*), tumor necrosis factor α (*tnf_α_*), immunoglobulin M heavy chain (*igm*), major histocompatibility complex II (*mhc-II*), interleukin 10 (*il-10*), caspase 3 (*casp*_3_) and prostaglandin E2 receptor *ep*_2_, *ep*_3_, and *ep*_4_ of gilthead sea bream juveniles fed the experimental diets at 4 h after the killed *Phdp* injection. Sample expressions were calibrated with the sample with a higher expression value and then normalized with elongation factor 1-alpha (*EF1α*) and ribosomal RNA 18S (*18S*) expressed transcripts. Values (means ± SE of the mean) presented as fold change of normalized expression level relative to the PBS-injected fish (defined as 1). Two-way ANOVA: NS: non-significant (*p* ≥ 0.05).

**Figure 3 animals-13-01770-f003:**
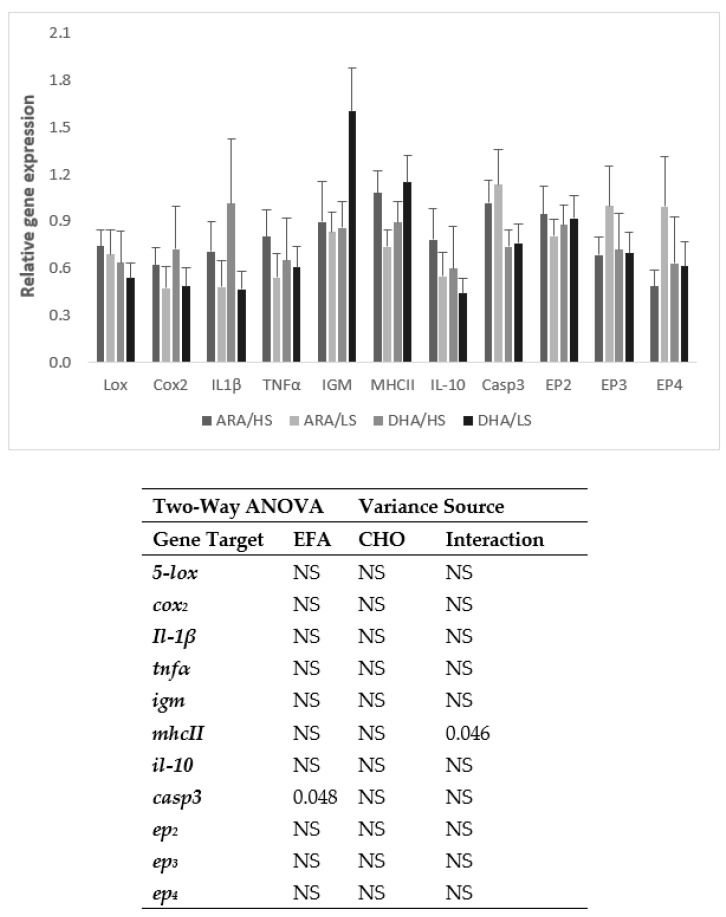
Intestinal expression levels of lipoxygenase (*5-lox*), cyclooxygenase 2 (*cox*_2_), interleukin 1β (*il-1_β_*), tumor necrosis factor α (*tnf_α_*), immunoglobulin M heavy chain (*igm*), major histocompatibility complex II (*mhc-II*), interleukin 10 (*il-10*), caspase 3 (*casp*_3_), and prostaglandin E2 receptor *ep*_2_, *ep*_3_, and *ep*_4_ of gilthead sea bream juveniles fed the experimental diets at 24 h after the killed *Phdp* injection. Sample expressions were calibrated with the sample with a higher expression value and then normalized with elongation factor 1-alpha (*EF1α*) and ribosomal RNA 18S (*18S*) expressed transcripts. Values (means ± SE of the mean; n = 6) presented as fold change of normalized expression level relative to the PBS-injected fish (defined as 1).Two-way ANOVA: NS: non-significant (*p* ≥ 0.05).

**Table 1 animals-13-01770-t001:** Proximate analysis of the experimental diets.

Diets	ARA/HS	ARA/LS	DHA/HS	DHA/LS
(ARA/DHA %)	2.3/0.3	2.3/0.3	0/2.6	0/2.5
Dry matter *(% DM)*	91.1	91.9	90.2	92.0
Crude Protein	46.5	46.7	46.9	46.1
Crude Lipid	17.6	17.7	18.7	17.7
Ash	8.6	9.2	8.9	8.7
Starch	19.6	4.6	20.0	5.4
Gross energy (kJ g^−1^)	23.4	23.0	23.1	23.1

CP: Crude protein; CL: Crude lipids; DM: Dry matter.

**Table 2 animals-13-01770-t002:** Essential fatty acid composition of the experimental diets (% of total fatty acids).

Diets	ARA/HS	ARA/LS	DHA/HS	DHA/LS
(ARA/DHA %)	2.3/0.3	2.3/0.3	0/2.6	0/2.5
18:2n-6 (LA)	15.80	15.55	15.25	15.13
18:3n-3 (ALA)	14.88	14.98	17.87	17.95
20:4n-6 (ARA)	13.11	13.24	0.17	0.14
20:5n-3 (EPA)	1.21	1.24	2.66	2.69
22:6n-3 (DHA)	1.53	1.60	13.67	13.90
∑SFA	24.20	24.12	19.30	19.06
∑MUFA	25.73	25.67	26.78	26.79
∑PUFA	49.25	49.38	53.44	53.67
∑n-6	30.86	30.76	15.53	15.37
∑n-3	18.10	18.32	37.63	38.00
n-6/n-3	1.70	1.68	0.41	0.40

LA: Linoleic acid; ALA: α-Linolenic acid; MUFA: monounsaturated fatty acid; PUFA: polyunsaturated fatty acid; SFA: saturated fatty acid.

**Table 3 animals-13-01770-t003:** Primer pair sequences used for the transcript level determination of selected genes.

Gene	Gene Abbreviation	Primer Sequences (5′ → 3′)	Primer Efficiency	Accession Number
Arachidonate 5-lipoxygenase	*5-Lox*	F: CCTGGCAGATGTGAACTTGA R: CGTTCTCCTGATACTGGCTGA	1.88	FP334124
Caspase 3	*Casp_3_*	F: CCAGTCAGTCGAGCAGATGA R: GAACACACCCTCGTCTCCAT	1.98	EU722334
Interleukin 1-β	*IL1β*	F: GGGCTGAACAACAGCACTCTC R: TTAACACTCTCCACCCTCCA	2.14	AJ277166
Immunoglobulin M	*IGM*	F: CAGCCTCGAGAAGTGGAAAC R: GAGGTTGACCAGGTTGGTGT	2.10	AM493677
Cyclooxygenase 2	*Cox* _2_	F: GAGTACTGGAAGCCGAGCAC R: GATATCACTGCCGCCTGAGT	1.96	AM296029
Tumor necrosis factor-α	*TNFα*	F: TCGTTCAGAGTCTCCTGCAG R: CATGGACTCTGAGTAGCGCGA	2.02	AJ413189
Major histocompatibility complex class IIa	*MHCII*	F: CTGGACCAAGAACGGAAAGA R: CATCCCAGATCCTGGTCAGT	1.94	DQ019401
Interleukin 10	*Il-10*	F: TGGAGGGCTTTCCTGTCAGA R: TGCTTCGTAGAAGTCTCGGATGT	1.94	FG261948
Prostaglandin E2 receptor EP2	*EP2*	F: ACACGTTGGACAGAGCAAGT R: TAGTGGCACGTGTCGTTCTC	2.12	XM_030410284.1
Prostaglandin E2 receptor EP3	*EP3*	F: ACCCCATGAAACCACTGTCC R: CCCTGGGCGGATACACAAAT	1.80	XM_030434233.1
Prostaglandin E2 receptor EP4	*EP4*	F: GCGGACAGACTCTCTGGTTC R: TCTCAGTGCTCAACGACACC	1.99	XM_030415999.1
Elongation factor 1α	*Ef1α*	F: CTGTCAAGGAAATCCGTCGT R: TGACCTGAGCGTTGAAGTTG	1.87	AF184170
Ribosomal protein S18	*18s*	F: AGGGTGTTGGCAGACGTTAC R: CTTCTGCCTGTTGAGGAACC	1.95	AM490061

**Table 4 animals-13-01770-t004:** Haematological profile of gilthead sea bream fed the experimental diets for 84 days.

Diets	ARA/HS	ARA/LS	DHA/HS	DHA/LS	PSE
(ARA/DHA %)	2.3/0.3	2.3/0.3	0/2.6	0/2.5	
RBC (×10^6^ μL^−1^)	2.7	3.1	3.1	3.3	0.07
Hemoglobin (g dL^−1^)	3.5	3.8	4.1	4.2	0.12
Hematocrit (%)	39.6	40.1	42.1	40.1	1.55
MCV (μm^3^)	146.2	132.9	133.4	122.4	3.44
MCH (pg cell^−1^)	13.1	12.7	13.2	12.0	0.35
MCHC (g 100 mL^−1^)	9.0	9.6	9.9	9.8	0.26
WBC (×10^4^ μL^−1^)	2.9	1.5	2.7	2.7	0.18
Neutrophils (10^4^ μL^−1^)	0.2	0.1	0.2	0.2	0.03
Monocytes (10^4^ μL^−1^)	0.1	0.1	0.2	0.1	0.01
Lymphocytes (10^4^ μL^−1^)	0.7	0.3	0.6	0.6	0.06
Thrombocytes (10^4^ μL^−1^)	1.8 ^B^	1.0 ^A^	1.7	1.7	0.11
** *Two-way ANOVA* **					
**Parameters**	**EFA**	**CHO**	**INT**		
RBC	0.039	NS	NS		
Hemoglobin	0.049	NS	NS		
Hematocrit	NS	NS	NS		
MCV	NS	NS	NS		
MCH	NS	NS	NS		
MCHC	NS	NS	NS		
WBC	NS	NS	NS		
Neutrophils	NS	NS	NS		
Monocytes	NS	NS	NS		
Lymphocytes	NS	NS	NS		
Thrombocytes	NS	NS	0.036		

Values are presented as means (n = 9) and pooled standard error (PSE). Two-way ANOVA: NS: non-significant (*p* ≥ 0.05). If the interaction was significant, one-way ANOVA was performed for each factor. Different capital letters indicate differences between CHO levels within each EFA. MCH: mean corpuscular haemoglobin; MCHC: mean corpuscular haemoglobin concentration; MCV: mean corpuscular volume; RBC: red blood cells; WBC: white blood cells.

**Table 5 animals-13-01770-t005:** Plasma immune parameters in gilthead sea bream fed the experimental diets for 84 days.

Diets	ARA/HS	ARA/LS	DHA/HS	DHA/LS	PSE
(ARA/DHA %)	2.3/0.3	2.3/0.3	0/2.6	0/2.5	
Nitric oxide (μM)	662.4	688.2	735.1	767.9	18.4
Total Ig (mg ml^−1^)	18.3	16.5	17.2	18.5	0.46
Peroxidase activity (units mL^−1^)	22.5	22.3	32.2	22.7	1.74
Protease activity (%)	5.7	5.6	5.1	5.3	0.09
Antiprotease activity (%)	82.7	81.3	84.4	83.3	0.39
Alternative complement pathway activity (ACH_50_ U mL^−1^)	2.48	1.81	4.43	4.46	0.35
Bactericidal activity against *Phdp (%)*	19.1	37.8	26.1	24.1	2.72
Bactericidal activity against *Vibrio anguillarum (%)*	^B^ 64.4	^A^ 51.6	60.8	61.5	1.62
** *Two-way ANOVA* **					
**Parameters**	**EFA**	**CHO**	**INT**		
Nitric oxide	0.040	NS	NS		
Total Ig	NS	NS	NS		
Peroxidase	NS	NS	NS		
Protease activity	0.004	NS	NS		
Antiprotease activity	0.015	NS	NS		
ACH_50_	0.001	NS	NS		
Bactericidal activity against *Phdp*	NS	NS	NS		
Bactericidal activity against *Vibrio anguillarum*	NS	0.047	0.028		

Values are presented as means (n = 9) and pooled standard error (PSE). Two-way ANOVA: ns: non-significant (*p* ≥ 0.05). If the interaction was significant, one-way ANOVA was performed for each factor. Different capital letters indicate differences between CHO levels within each EFA.

**Table 6 animals-13-01770-t006:** Hematological profile of gilthead sea bream at 4 h after challenge with killed *Phdp*.

Diets	ARA/HS	ARA/LS	DHA/HS	DHA/LS	PSE
(ARA/DHA %)	2.3/0.3	2.3/0.3	0/2.6	0/2.5	
RBC	0.77 _a_	0.86	1.16 _b_	0.86	0.05
Hemoglobin	1.00	1.07	1.06	0.94	0.03
Hematocrit	0.85	0.95	1.11	1.01	0.03
MCV	1.08	1.11	0.93	1.11	0.03
MCH	1.19	1.26	0.97	0.96	0.07
MCHC	1.05	1.07	0.95	0.93	0.02
WBC	^B^ 1.25	^A^ 0.86 _a_	1.35	1.60 _b_	0.09
Neutrophils	1.19	1.05	1.35	2.06	0.14
Monocytes	1.62	0.90	0.78	0.72	0.15
Lymphocytes	^B^ 1.24	^A^ 0.67	0.78	0.94	0.08
Thrombocytes	1.26	0.81	1.34	1.60	0.11
** *Two-way ANOVA* **					
**Parameters**	**EFA**	**CHO**	**INT**		
RBC	0.024	NS	0.018		
Hemoglobin	NS	NS	NS		
Hematocrit	0.002	NS	NS		
MCV	NS	NS	NS		
MCH	NS	NS	NS		
MCHC	0.019	NS	NS		
WBC	0.007	NS	0.040		
Neutrophils	NS	NS	NS		
Monocytes	NS	NS	NS		
Lymphocytes	NS	NS	0.040		
Thrombocytes	0.033	NS	NS		

Values are presented as means (n = 6) and pooled standard error and represented as fold change relative to the PBS-injected fish (defined as 1). MCH: mean corpuscular hemoglobin; MCHC: mean corpuscular hemoglobin concentration; MCV: mean corpuscular volume; RBC: red blood cells; WBC: white blood cells. Two-way ANOVA: ns: non-significant (*p* ≥ 0.05). If the interaction was significant, one-way ANOVA was performed for each factor. Different capital letters indicate differences between CHO levels within each EFA and different subscript letters indicate differences between EFA within each CHO level.

**Table 7 animals-13-01770-t007:** Plasma immune parameters in gilthead sea bream juveniles at 4 h after challenge with killed *Phdp*.

Diets	ARA/HS	ARA/LS	DHA/HS	DHA/LS	PSE	
(ARA/DHA %)	2.3/0.3	2.3/0.3	0/2.6	0/2.5		
Nitric oxide	1.06	1.02	1.05	0.88	0.04	
Total Ig	1.46	1.92	0.92	0.96	0.16	
Peroxidase	1.65	0.53	1.54	1.50	0.24	
Protease activity	0.98	0.95	0.98	0.99	0.03	
Antiprotease activity	0.99	1.03	1.00	0.99	0.01	
ACH_50_	^B^ 1.40 _b_	^A^ 0.29 _a_	0.92 _a_	1.02 _b_	0.13	
Bactericidal activity against *Phdp*	0.64	0.54	0.89	0.46	0.10	
Bactericidal activity against *vibrio anguillarum*	1.26	1.11	0.97	1.01	0.04	
** *Two-way ANOVA* **						
**Parameters**	**EFA**	**CHO**	**INT**			
Nitric oxide	NS	NS	NS			
Total Ig	0.034	NS	NS			
Peroxidase	NS	NS	NS			
Protease activity	NS	NS	NS			
Antiprotease activity	NS	NS	NS			
ACH_50_	NS	0.013	0.005			
Bactericidal activity against *Phdp*	NS	NS	NS			
Bactericidal activity against *Vibrio anguillarum*	0.027	NS	NS			

Values are presented as means (n = 6) and pooled standard error and represented as fold change relative to the PBS-injected fish (defined as 1). Two-way ANOVA: ns: non-significant (*p* ≥ 0.05). If the interaction was significant, one-way ANOVA was performed for each factor. Different capital letters indicate differences between CHO levels within each EFA and different subscript letters indicate differences between EFA within each CHO level.

**Table 8 animals-13-01770-t008:** Hematological profile of gilthead sea bream at 24 h after challenge with killed *Phdp*.

Diets	ARA/HS	ARA/LS	DHA/HS	DHA/LS	PSE
(ARA/DHA %)	2.3/0.3	2.3/0.3	0/2.6	0/2.5	
RBC	0.93	0.88	0.92	0.75	0.03
Hemoglobin	0.89	1.03	0.96	0.96	0.02
Hematocrit	0.91	0.91 _a_	^A^ 0.89	^B^ 1.10 _b_	0.03
MCV	0.96	1.06	0.93	1.35	0.05
MCH	0.94	1.24	0.84	1.05	0.06
MCHC	^A^ 0.97 _a_	^B^ 1.16 _b_	^B^ 1.11 _b_	^A^ 0.86 _a_	0.03
WBC	1.48	1.49	0.96	0.90	0.10
Neutrophils	2.11	2.03	2.42	1.98	0.21
Monocytes	1.56	1.39	1.39	0.86	0.28
Lymphocytes	0.96	0.74	1.06	0.93	0.12
Thrombocytes	1.40	1.42	0.80	0.97	0.09
** *Two-way ANOVA* **					
**Parameters**	**EFA**	**CHO**	**INT**		
RBC	0.041	0.031	NS		
Hemoglobin	NS	0.044	NS		
Hematocrit	NS	0.032	0.032		
MCV	NS	0.004	NS		
MCH	NS	NS	NS		
MCHC	NS	NS	0.000		
WBC	0.004	NS	NS		
Neutrophils	NS	NS	NS		
Monocytes	NS	NS	NS		
Lymphocytes	NS	NS	NS		
Thrombocytes	0.004	NS	NS		

Values are presented as means (n = 6) and pooled standard error and represented as fold change relative to the PBS-injected fish (defined as 1). MCH: mean corpuscular hemoglobin; MCHC: mean corpuscular hemoglobin concentration; MCV: mean corpuscular volume; RBC: red blood cells; WBC: white blood cells. Two-way ANOVA: ns: non-significant (*p* ≥ 0.05). If the interaction was significant, one-way ANOVA was performed for each factor. A different capital letter indicates differences between CHO levels within each EFA and different subscript letters indicate differences between EFA within each CHO level.

**Table 9 animals-13-01770-t009:** Plasma immune parameters in gilthead sea bream juveniles at 24 h after challenge with killed *Phdp*.

Diets	ARA/HS	ARA/LS	DHA/HS	DHA/LS	PSE
(ARA/DHA %)	2.3/0.3	2.3/0.3	0/2.6	0/2.5	
Nitric oxide	1.26	0.97	1.11	0.92	0.05
Total Ig	2.22	1.17	0.77	0.62	0.18
Peroxidase	2.17	2.27	0.86	0.86	0.27
Protease activity	0.89	1.05	0.94	0.98	0.02
Antiprotease activity	0.99	1.01	0.99	1.00	0.00
ACH_50_	1.12	0.87	0.64	0.56	0.10
Bactericidal activity against *Phdp*	1.03	0.56	1.05	0.66	0.08
Bactericidal activity against *Vibrio anguillarum*	0.81	0.99	0.82	1.01	0.03
** *Two-way ANOVA* **					
**Parameters**	**EFA**	**CHO**	**INT**		
Nitric oxide	NS	0.017	NS		
Total Ig	0.002	NS	NS		
Peroxidase	0.012	NS	NS		
Protease activity	NS	0.020	NS		
Antiprotease activity	NS	0.006	NS		
ACH_50_	NS	NS	NS		
Bactericidal activity against *Phdp*	NS	0.010	NS		
Bactericidal activity against *Vibrio* *anguillarum*	NS	0.000	NS		

Values are presented as means (n = 6) and pooled standard error and represented as fold change relative to the PBS-injected fish (defined as 1). Two-way ANOVA: ns: non-significant (*p* ≥ 0.05). If the interaction was significant, one-way ANOVA was performed for each factor.

## Data Availability

The data presented in this study are available on request from the corresponding author.

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
