# Peer review of "Dietary ARA, DHA, and Carbohydrate Ratios Affect the Immune Status of Gilthead Sea Bream Juveniles upon Bacterial Challenge"

_animals, 2023, doi:10.3390/ani13111770_

Round 1

Reviewer 1 Report

To the authors, I congratulate the contextualization and complexity of the work, which is extremely relevant to promote European aquaculture. The draft is very well thought out with complementary techniques that are closed in an understanding of the context of the immunoresponse in the face of lipid and carbohydrate nutrition for gilthead sea bream juveniles. Minor revisions are suggested, mindful of the fact that they could provide growth data as the draft is based. During the discussion, one of the hypotheses is that n-3 LC-PUFA EFA could have affected the immune response, despite not having affected growth.

1.      Introduction

L 47-52: “marine fish are not able to bioconvert linoleic (18:2n-6) and α-linolenic (18:3n-3) acids into long-chain polyunsaturated fatty acids (LC-PUFA), such as arachidonic (ARA; 20:4n-6) and docosahexaenoic (DHA; 22:6n-3) acids due to the absence or low activity of ∆5-desaturase 51 and elongases (Elovl2)”.

Be very careful with generalizations, we have examples of marine fish that can biosynthesize LC-PUFAs by alternative routes, example rabbitfish (Siganus canaliculatus).

https://link.springer.com/article/10.1007/s10126-018-9854-0

https://www.sciencedirect.com/science/article/pii/S1096495917300052?casa_token=mw4R_eu540YAAAAA:lzHoKbBTKTch-G7K3ZtC1QB8a7M1PP2pHr_OsT3QQj0Y0FdSjj3Dxf2TdQb9YHAowGLGmivtlg

https://www.sciencedirect.com/science/article/pii/S0044848619333885?casa_token=HcmjRxdcmJoAAAAA:xUmExh7f4ahb1TxDQTRjudwX25tqLhrGKDzG2NPJklkm4upnzPsbw7i1yUV9JFgWi8yH5PutAg

https://onlinelibrary.wiley.com/doi/abs/10.1111/anu.12178?casa_token=CwL4MFVLOs4AAAAA:Ayv_DuY_0jJKdoeh4jfk19IukQai9QYWmQ1Jq6m0eQKVSEIsznD-x121gvX9_8D1cf2PG3jUTGm5qA

L 53-67 - I believe that the second and third paragraphs can be merged

L 74-75 - Displaced paragraph, can be merged with the previous paragraph.

L77 - “In gilthead sea bream, several studies have already evaluated the effects of different dietary EFA profiles on the immunological status of fish under unchallenged and challenging conditions”

I believe that at this point I could return to the null capacity of biosynthesis of LCPUFA by the species (with sitations), the initial idea of the introduction.

2.      Results

L 247 “The growth performance of the fish fed the experimental diets was not the aim of this study and results were presented elsewhere”

I believe it is not the most elegant way to start the results, already with a denial/negation, even more with relevant information on growth performance, in fact I do not understand why the data on growth performance was not presented in this draft (at least in the complementary).

3.      Discussion

L396: seems to have an extra space before “n-3 LC-PUFA EFA deficiency”

L404: “Although this amount seemed sufficient to promote an adequate growth performance” - highlights the importance of growth data

4.      Conclusions

would it be possible to indicate for future studies some nutritional modulation that regulates positively (more accentuated) the expression of immune-related genes facing a challenge?

Author Response

To the authors, I congratulate the contextualization and complexity of the work, which is extremely relevant to promote European aquaculture. The draft is very well thought out with complementary techniques that are closed in an understanding of the context of the immunoresponse in the face of lipid and carbohydrate nutrition for gilthead sea bream juveniles. Minor revisions are suggested, mindful of the fact that they could provide growth data as the draft is based. During the discussion, one of the hypotheses is that n-3 LC-PUFA EFA could have affected the immune response, despite not having affected growth.

  1. Introduction

L 47-52: “marine fish are not able to bioconvert linoleic (18:2n-6) and α-linolenic (18:3n-3) acids into long-chain polyunsaturated fatty acids (LC-PUFA), such as arachidonic (ARA; 20:4n-6) and docosahexaenoic (DHA; 22:6n-3) acids due to the absence or low activity of ∆5-desaturase 51 and elongases (Elovl2)”.

Be very careful with generalizations, we have examples of marine fish that can biosynthesize LC-PUFAs by alternative routes, example rabbitfish (Siganus canaliculatus).

https://link.springer.com/article/10.1007/s10126-018-9854-0

https://www.sciencedirect.com/science/article/pii/S1096495917300052?casa_token=mw4R_eu540YAAAAA:lzHoKbBTKTch-G7K3ZtC1QB8a7M1PP2pHr_OsT3QQj0Y0FdSjj3Dxf2TdQb9YHAowGLGmivtlg

https://www.sciencedirect.com/science/article/pii/S0044848619333885?casa_token=HcmjRxdcmJoAAAAA:xUmExh7f4ahb1TxDQTRjudwX25tqLhrGKDzG2NPJklkm4upnzPsbw7i1yUV9JFgWi8yH5PutAg

https://onlinelibrary.wiley.com/doi/abs/10.1111/anu.12178?casa_token=CwL4MFVLOs4AAAAA:Ayv_DuY_0jJKdoeh4jfk19IukQai9QYWmQ1Jq6m0eQKVSEIsznD-x121gvX9_8D1cf2PG3jUTGm5qA

R: The text was modified to accommodate the reviewer concerns.

L 53-67 - I believe that the second and third paragraphs can be merged

R: Done.

L 74-75 - Displaced paragraph, can be merged with the previous paragraph.

R: Done.

L77 - “In gilthead sea bream, several studies have already evaluated the effects of different dietary EFA profiles on the immunological status of fish under unchallenged and challenging conditions”

I believe that at this point I could return to the null capacity of biosynthesis of LCPUFA by the species (with sitations), the initial idea of the introduction.

R: The text was modified accordingly.

  1. Results

L 247 “The growth performance of the fish fed the experimental diets was not the aim of this study and results were presented elsewhere”

I believe it is not the most elegant way to start the results, already with a denial/negation, even more with relevant information on growth performance, in fact I do not understand why the data on growth performance was not presented in this draft (at least in the complementary).

R: A brief description of the growth trial results is presented in lines 256-260. However, the growth results are presented in a previously published paper in Magalhães, et al. [1]

 Magalhães, R.; Martins, N.; Fontinha, F.; Moutinho, S.; Olsen, R.E.; Peres, H.; Oliva-Teles, A. Effects of dietary arachidonic acid and docosahexanoic acid at different carbohydrates levels on gilthead sea bream growth performance and intermediary metabolism. Aquaculture 2021, 545, 737233, doi:https://doi.org/10.1016/j.aquaculture.2021.737233

  1. Discussion

L396: seems to have an extra space before “n-3 LC-PUFA EFA deficiency”

R: Done

L404: “Although this amount seemed sufficient to promote an adequate growth performance” - highlights the importance of growth data

R: R: The growth results are presented in lines 256-260 and in a previously published paper Magalhães, et al. [1]

 Magalhães, R.; Martins, N.; Fontinha, F.; Moutinho, S.; Olsen, R.E.; Peres, H.; Oliva-Teles, A. Effects of dietary arachidonic acid and docosahexanoic acid at different carbohydrates levels on gilthead sea bream growth performance and intermediary metabolism. Aquaculture 2021, 545, 737233, doi:https://doi.org/10.1016/j.aquaculture.2021.737233

  1. Conclusions

would it be possible to indicate for future studies some nutritional modulation that regulates positively (more accentuated) the expression of immune-related genes facing a challenge?

R: A phrase was added to the conclusion regarding this subject.

Reviewer 2 Report

All the comments can be found in the attached file.

Author Response

I have had the opportunity to review the manuscript entitled “Dietary ARA, DHA, and carbohydrate

ratios affect the immune status of gilthead sea bream juveniles upon bacterial challenge” (Ref. no.

animals-2360075).

Authors have explored the effects of different dietary n-6/n-3 LC-PUFA ratios and carbohydrate

content in the immune response of gilthead seabream before and after being submitted to a

bacterial challenge.

Overall, this is an interesting study as both reducing fish oil and increasing carbohydrate levels in

feed have been common practices in aquaculture for productive purposes but little attention has

been paid to their implications on animal health and welfare.

However, a major drawback and corrections have to be addressed. Each comment and/or correction listed below is highlighted on the marked copy of the MS that can be found in this document after the comments.

General:

-The major drawback that needs to be addressed concerns the discussion. The apparent beneficial effect of CHO on plasma immune response is poorly discussed. Although the information available in this regard in fish is very scarce, there are some studies that are not referenced in the MS that I think can help to perform a more in-depth analysis of the results: https://doi.org/10.1111/j.1365-2761.1994.tb00220.x; https://doi.org/10.1016/j.aqrep.2020.100515; https://doi.org/10.1155/2022/7820017; https://doi.org/10.1038/s41598-021-86172-8;

Also this article with mice can be useful: https://doi.org/10.1016/j.isci.2021.102835

I think these articles can help develop a more robust discussion and, accordingly, the simple

summary, abstract, introduction, and conclusion sections would need to be modified.

R: Done accordingly.

-Results: standard error of the mean (SEM) and pooled standard error (SE) have been confused.

What appears in the tables is the pooled standard error (SE) and error bars in the figures correspond to the standard error of mean. So, SEM has to be changed to SE in all tables and the footnote in tables and figures corrected. See below.

R: To accommodate the reviewer concern the authors changed standard error of the mean (SEM) for pooled standard error (PSE), and in figures SE of each mean was added.

Other general comments:

-Avoid the use of abbreviations when they have not been previously defined; eg. Ln 13: LC-PUFA.

R: Done.

-References section: An error has occurred in listing two references: 35 is 34 and vice versa.

R: Done.

Corrections needed in text are listed below.

Additional comments (highlighted in the marked copy of the MS):

Abstract

Ln. 21: this study aims to assess

R:Done

Ln. 21: LC-PUFA

R:Done

Lns. 24-25: Replace with “diets with high (20%) or low (5%) level of gelatinized starch (HS and LS

diets, respectively) and including approximately 2.4 % ARA or DHA.”

R: Done.

Ln. 33: ARA/LS

R: Done.

Ln. 37: protease and antiprotease activivity,

R: Done.

2

Introduction

Ln. 5: add comma

R: Done.

Ln. 53: properties.

R: Done.

Ln. 57: define both abbreviations

R: Done.

Ln. 69: define both abbreviations

R: Done.

Ln. 57: define abbreviation

R: Done.

Materials and methods

Ln. 110: et al.

R: Done.

Ln. 142-143: scientific name in italics; subspecies in lowercase

R: Done.

Ln. 143: delete comma after family name; reference number 35

R: Done.

Ln. 150: italics

R: Done.

Ln. 158: reference number 34

R: Done.

Ln. 165: groups

R: Done.

Ln. 184: delete comma after family name; “et al.” in regular font style

R: Done.

Ln. 186: italics

R: Done.

Ln. 198: delete comma after family name; “et al.” in regular font style

R: Done.

Ln. 218: delete comma after family name; “et al.” in regular font style

R: Done.

Ln. 223: al. (insert dot); reference number 34

R: Done.

Ln. 228: receptors

R: Done.

Ln. 234: delete comma after family name; “et al.” in regular font style

R: Done.

Ln. 235; Table 3: marked words in lowercase

R: Done.

Ln. 239: means and pooled standard error (SE)

R: Done.

Ln. 241: delete the highlighted text

R: Done.

Ln. 248: reference number 34

R: Done.

Ln. 258: ARA/LS

R: Done.

Ln. 269; Table 4: correct SEM by SE

R: Done.

Ln. 260: pooled standard error.

R: Done.

Ln. 265: antiprotease

R: Done.

Ln. 269; Table 5:

 -correct SEM by SE

R: Done.

 -no significant differences between highlighted data?

R: the authors double checked, and no differences were found.

Ln. 270: pooled standard error.

R: Done.

Ln.274: lowercase

R: Done.

Lns. 281-283: all marked words in lowercase

R: Done.

Ln. 283: insert a comma after (casp3)

R: Done.

Ln. 283: receptors

R: Done.

3

Ln. 287: mean (n=9) ± standard error of the mean (SEM);

R: Done.

Ln. 295; Table 6: correct SEM by SE

R: Done.

Ln. 296: Values, presented as means (n=9) and pooled standard error, are expressed as fold change…

R: Done.

Ln. 303: Ig

R: Done.

Ln. 308; Table 7:

-correct SEM by SE

R: Done.

-Vibrio (capitalized)~

R: Done.

Ln. 310: Values, presented as means (n=9) and pooled standard error, are expressed as fold change…

R: Done.

Ln. 315: intestine

R: Done.

Lns. 321-323: all marked words in lowercase

R: Done.

Ln. 323: insert a comma after (casp3)

R: Done.

Ln. 324: receptors

R: Done.

Ln. 327: (mean ± SEM; n=9);

R: Done.

Ln. 335; Table 8: correct SEM by SE

R: Done.

Ln. 336: Values, presented as means (n=9) and pooled standard error, are expressed as fold change…

R: Done.

Ln. 350; Table 9:

-correct SEM by SE

R: Done.

-Vibrio (capitalized)´

R: Done.

-anguillarum

R: Done.

Ln. 352: Values, presented as means (n=9) and pooled standard error, are expressed as fold change…

R: Done.

Lns. 372-374: all marked words in lowercase

R: Done.

Ln. 374: insert a comma after (casp3)

R: Done.

Ln. 375: receptors

R: Done.

Ln. 378: (mean ± SEM; n=9)

R: Done.

Discussion

Ln. 467: COX2 and 5-LOX

R: Done.

Ln. 469: both in lowercase

R: Done.

Conclusion

Ln. 488: shows

R: Done.

References

-Use only one way to express the doi:

-“doi: 10….” Or http://….

- some references showed the two options (highkighted)

R: Done.

-References without doi: 3, 6, 9, 11, 12, 13, 15, 16, 17, 21, 25, 26, 30, 31, 32, 33, 34, 36, 37, 41, 43,

45, 46.

R: Done.

-According to the instructions for authors, journal name in references should be abbreviated. Thereis a high number of references with the full name.

R: Done.

Ln. 561: italics

R: Done.

Ln. 562: information regarding volume and pages is missing: (53, 6007-6019)

R: Done.

Ln. 572: lower case

R: Done.

Ln. 588: all words in the title are capitalized, change to “Sentence case”

R: Done.

Ln. 590: all words in the title are capitalized, change to “Sentence case”

R: Done.

Ln. 596: Díaz-Rosales et al., is the reference 35

R: Done.

Ln. 598: Magalhães et al. is the reference 34

R: Done.

Ln. 614: The names of the editors are repeated

R: Done.

Ln. 616:

- all words in the title are capitalized, change to “Sentence case”

-correct scientific name of sea bream

R: Done.

Reviewer 3 Report

Could you replace growth trial by feeding trail along the MS, where the study did not show growth results and focused on the feeding effect of immune responses and molecular.

L28: revise the grammars “presented increased” also make a double check on the whole documents.

Remove the space before % along the MS.

L142: italicized the Latin names “P.  damselae”, consider along the MS.

The formula of MCH (pg cell-1) = (HH/RBC) x 10 need to be revised. Especially the “HH” abbreviation.

In all tables dose the treatment name reported correctly as abbreviation, but as a percent is not clear (high starch (20%) low starch (5%), also ARE and DHA percent it’s not clear).

Diets

ARA/HS

ARA/LS

DHA/HS

DHA/LS

(ARA/DHA ratio)

2.3/0.3

2.3/0.3

0/2.6

0/2.5

L 222: what is the meaning of DI, please revised all abbreviations along the MS

Figure 1, 2, and 3 B is not figures convert to tables.

make a double check on the whole documents grammar and bunctiuations.

Author Response

Comments and Suggestions for Authors

Could you replace growth trial by feeding trail along the MS, where the study did not show growth results and focused on the feeding effect of immune responses and molecular.

R: Done.

L28: revise the grammars “presented increased” also make a double check on the whole documents.

R: Done.

Remove the space before % along the MS.

R: Done.

L142: italicized the Latin names “P.  damselae”, consider along the MS.

R: Done.

The formula of MCH (pg cell-1) = (HH/RBC) x 10 need to be revised. Especially the “HH” abbreviation.

R: Done.

In all tables dose the treatment name reported correctly as abbreviation, but as a percent is not clear (high starch (20%) low starch (5%), also ARE and DHA percent it’s not clear).

Diets

ARA/HS

ARA/LS

DHA/HS

DHA/LS

(ARA/DHA ratio)

2.3/0.3

2.3/0.3

0/2.6

0/2.5

R: The word ratio was replaced by percentage in the tables for readability purposes.

L 222: what is the meaning of DI, please revised all abbreviations along the MS

R: Done accordingly, the meaning of DI is explained at line 166.

Figure 1, 2, and 3 B is not figures convert to tables.

R: A and B together represented the figure 1, 2 and 3. To avoid confusion changes were made regarding the reviewer concern.

 Q: Comments on the Quality of English Language make a double check on the whole documents grammar and bunctiuations.

R: Done accordingly.

Round 2

Reviewer 2 Report

All my comments are in the attached file.

Author Response

 The authors have corrected most of the comments in my previous report, related to minor text editing corrections, but unfortunately have ignored the main drawback that needed to be addressed: to discuss the results about the role of CHO on the immune response. They have merely related the results of two of the references I mentioned in my previous report, but the main question remains unresolved: Why does the increase in dietary carbohydrates improve the immune system’s responsiveness? Since all diets were isoenergetic, it seems evident that macronutrient composition rather than energy was affecting immune response. I insist that a slow and comprehensive reading of the reference https://doi.org/10.1016/j.isci.2021.102835 provides information that will allow the authors to discuss these results, which so far has not been done.

R: The authors made modifications in the introduction and discussion to better frame the subject and discuss the results. We hope that these modifications satisfied the reviewer's concerns.

Additional comments:
In table 5, for each parameter the units of expression are provided. This information is not included in tables 7 and 9.

R: We thank the reviewer's suggestion but the authors do not include this information in Tables 7 and 9 because in these tables the data are presented as fold change and in Table 5 are not.

Other comments (highlighted in red on the marked copy of the MS that can be found below):
Ln. 33: Ig

R: Done.
Ln. 49: ..a great deal of because. “
attention” is missing after “of”

R: Done.
Ln. 114: Magalhães et. [37]. “et
al.

R: Done.
Ln. 251: detele “diets”

R: Done.
LN. 272. Table 5: (the same in tables 7 and 9)

R: Done.
-Units: m
L-1

R: Done.
-Peroxidase activity

R: Done.
-Protease activity

R: Done.

-Antiprotease activity

R: Done.
Ln. 356. Table 9. “
anguillarum

R: Done.
Ln. 450: activities

R: Done.
Ln. 461: haemolitic

R: Done.
Ln. 463: “reduced”

R: Done.
Ln. 464: “activity”

R: Done.
Ln. 466: “dietary”

R: Done.
Ln. 505: delete comma

R: Done.
Ln. 505: to

R: Done.